# Integrated transcriptome and metabolome analysis of melanin-related genes and metabolites in Muchuan Black Bones Chickens

Liu Wei[1], Jie Hong-Wei[1], Gao Ming-Chao[2], Yao Xiu-Mei[1], Yin Qiong[1], Xiang Hai[1], Ma Zheng[1], Zhang Zheng-Fen[2], Li Liang[3], Li Hua[2], Qi Hao[1]*, Ye Fei[1]*

1 Guangdong Provincial Key Laboratory of Animal Molecular Design and Precise Breeding, School of Animal Science and Technology, Foshan University, Foshan, China, 2 Guangdong Tinoo's Food Co., Ltd., Qingyuan, China, 3 Animal Husbandry and Veterinary Institute of Guizhou Province, Guizhou, China

* q15663857127@163.com (QH); yefei0831@fosu.edu.cn (YF)

## Abstract

The Muchuan Black Bones Chickens represent a significant subtype of the Sichuan mountain black bones chickens. In the breeding of black bone chickens, the degree of melanization tends to separate due to hybridization, while molecular-assisted breeding techniques were still limited, and the key genes involved in melanin deposition in the early growth of black bones chickens have yet to be validated. Hypothesize that melanin deposition in Muchuan Black Bones Chickens was governed by specific genetic and metabolic regulators that can be systematically identified. This study aims to identify key genes and metabolites associated with melanin in Muchuan Black Bones Chickens through integrated transcriptomic and metabolomic analyses. CDP-ethanolamine, m-trigallic acid, UDP-4-dehydro-6-deoxy-D-glucose, glucoiberverin, and uridine diphosphate glucuronic acid (UDP glucuronic acid) were identified as significant differential metabolites, which were intimately linked to phenylalanine, tyrosine, and tryptophan biosynthesis associated with melanin accumulation. A total of 73 differentially expressed genes, including *cAMP response element-binding protein 5 (CREB5)* and *GIPC PDZ domain containing family member 2 (GIPC2),* were enriched in the neuroactive ligand-receptor interaction pathway, arginine-proline metabolism pathway, histidine metabolism, terpenoid backbone biosynthesis, and glycerophospholipid metabolism. This study identified significant metabolites and regulatory genes associated with melanin synthesis, establishing a theoretical foundation for the breeding and selection of chickens.

## Introduction

Muchuan Black Bones Chickens, also referred to as "West Sichuan Black Bones Chickens" [1], were renowned for their distinctive black bones and rich nutritional value [2–4]; however, they exhibit segregation in melanin deposition, which emerged

**Data availability statement:** Data are available via the CNGB Sequence Archive (CNSA) of China National GeneBank DataBase (CNGBdb) (https://db.cngb.org/data_resources/?query=CNP0006904).

**Funding:** This research was funded by the Guangdong Basic and Applied Basic Research Foundation (2022A1515111078, awarded to YF) and Guizhou Science and Technology Key Project ([2022] No. 034, awarded to LL). The funders had no role in study design, data collection and analysis, decision to publish, or preparation of the manuscript.

**Competing interests:** The authors have declared that no competing interests exist.

in the early stages of growth. Melanin can be divided into five different types: eumelanin, fusomelanin, isomelanin, neuromelanin, and pyomelanin [5]. Black bone chickens primarily consist of eumelanin, a polymer made from 5, 6-dihydroxyindole (DHI) and 5, 6-dihydroxyindole-2-carboxylic acid (DHICA) [6]. The synthesis and transport of melanin were regulated by multiple genes, such as *Endothelin Receptor Type B2* (*EDNRB2*) [7], *Endothelin 3* (*EDN3*) [8], *tyrosinase* (*TYR*) [9], *tyrosinase-related protein 1 (TYRP1)* [10]*, microphthalmia-associated transcription factor* (*MITF*) [11], and *melanosome protein 17* (*PMEL17*) [12]. *EDNRB2*, a G protein-coupled receptor, activates downstream signaling pathways upon binding with *EDN3,* which supports the migration of neural crest-derived melanoblasts to skin and feather regions. The *TYR* and *TYRP1* families formed an enzymatic network for melanin synthesis, where *TYR* catalyzed the oxidation of tyrosine to dopaquinone, while *TYRP1* stabilized *TYR* enzyme activity, catalyzed the isomerization of dopa pigment, and facilitated deposition onto the melanosome matrix fiber formed by *PMEL17*. The *MLPH* gene dynamically regulated melanin from synthesis to distribution, whereas *MITF* influenced the expression of *MLPH* and indirectly affected melanin formation. Furthermore, the interaction between *MITF* and the M-box motif is essential for the coordinated transcriptional regulation of tyrosinase family genes, highlighting the pivotal role of *MITF* in controlling melanin biosynthesis. Previously, many molecular biological methods studying the melanin deposition included transcriptomic and proteomic [13–14], SNP molecular marker screening [15]. Melanin content was determined using spectrophotometry [16], high-performance liquid chromatography (HPLC) [17], and fluorescence analysis [18], while melanin extraction was achieved through enzymolysis [19], a three-step method (degreasing-enzymolysis-acidolysis) [20].

At the early growth stage of black bones chickens, melanin is deposited in the breast and skin with a marked degree of separation. It is hypothesized that breast muscle melanin deposition in Muchuan Black Bones Chickens is regulated by specific genetic and metabolic factors. To identify candidate molecular markers connected to melanin deposition in the early development phase of Muchuan Black Bones Chickens, this study utilized 35-day-old Muchuan Black Bones Chickens as experimental subjects to identify key genetic markers and metabolites associated with early breast muscle melanin deposition.

## Materials and methods

### Sample collection

All Muchuan Black Bones Chickens were sourced from Muchuan County Black Phoenix Black Bone Chickens Industry Co., Ltd. The chickens were raised at the Foshan Animal Breeding Engineering Technology Research Center, following standard broiler feeding protocols until 35 days of age. A total of 60 chickens aged 35 days old were selected randomly. Euthanasia was carried out by using the gas fainting method. Gently place the poultry into a sealed chamber, close the lid securely, and gradually introduce $CO_2$. The initial gas concentration should be maintained at 30% to 50%. Following confirmation of unconsciousness, increase the $CO_2$ concentration to at

least 90%, and continue to maintain this environment for another 5–10 minutes. Once death has been confirmed, proceed with sample collection. Whole blood (2 ml) was collected via wing veins and stored in collection vessels at −20°C for sequencing purposes. Breast muscle was also collected and stored at −80°C for subsequent analysis.

## Determination of melanin in breast muscle

The melanin concentration in the breast muscles was determined through a chicken melanin double-antibody sandwich enzyme-linked immunosorbent assay (ELISA) kit (manufacturer: Shanghai Enzyme Biotech Co., Ltd.; model: YJ500392) [21]. A total of 0.1 g of breast muscle was homogenized in 900 μL PBS buffer, and the supernatant was collected. Standard wells, blank control wells, and sample wells were assigned on the ELISA plate prior to the assay. Three technical replicates were established for each condition, with corresponding reagents and samples added according to the manufacturer's instructions. Following incubation and washing steps, a termination solution was added, and the OD value (absorbance) of each well was measured at a wavelength of 450 nm using an ELISA reader, with results recorded within 15 minutes [22].

## RNA-seq sequencing

Samples were categorized into a high-melanin group (MH) and a low-melanin group (ML) based on the measured melanin content of the breast muscle. Five chickens from each group were selected, and appropriate amounts of breast muscle tissue samples were collected for mRNA transcriptome sequencing. To confirm the reliability of transcriptome sequencing data, the extracted total RNA samples were subjected to rigorous quality control assessments, including purity evaluation, concentration measurement, and integrity verification [23]. The sample concentrations ranged between 420 and 740 ng/μL, with OD 260/280 values [24] approximately around 2.0 by using NanoDrop 2000 spectrophotometer, and all integrity evaluation index RIN values from 7 to 7.6 by using the Agilent 2100/LabChip GX [25]. All total RNAs met quality standards and were suitable for subsequent experimental steps. The library was constructed, including mRNA interruption and enrichment, cDNA purification, end repair, A-tail and splicing connected for sequencing, fragment size selection with AMPure XP beads, and a cDNA library obtained by PCR enrichment. After the library construction, the quality control was performed on the resulting libraries. After passing quality inspection, the transcriptome library was sequenced in PE150 mode using the Illumina NovaSeq 6000 sequencing platform. After the sequencing data were obtained, BMKCloud (https://international.biocloud.net/zh/dashboard) was used to analyze differentially expressed genes through the DESeq2-edgeR methods, with an FDR threshold of 0.05 and a fold change (FC) of 1.5 as the criterion for significance. Differentially expressed genes were analyzed for functional enrichment using the DAVID Bioinformatics Resources database (https://david.ncifcrf.gov/), including pathway enrichment analyses from the Gene Ontology (GO) and the Kyoto Encyclopedia of Genes and Genomes (KEGG). Additionally, Sangerbox 3.0 (http://www.sangerbox.com/) was used to visualize genetic variations via volcano plots and to generate GO and KEGG bubble charts.

## Assessment via quantitative real-time PCR (RT-qPCR)

Reverse transcription was performed with HiScript III All-in-one RT SuperMix (Vazyme, China), yielding cDNA at concentrations≥1 ng/μL. Primer sequences were designed using the NCBI's Primer–BLAST platform. The primer sequences were shown in Table 1. qPCR was carried out with SYBR Green Master Mix (Vazyme) to determine relative mRNA expression. The Qubit 3.0 (Thermo) fluorometer was used. The results were analyzed based on calculation $2^{-\Delta\Delta Ct}$.

## Extraction and detection of metabolites

Metabolomic analysis was performed through a standardized protocol involving initial metabolite extraction, instrumental detection, and final metabolite identification and quantification. Firstly, six chickens in the ML group and six chickens

**Table 1. Differential gene primer sequence.**

| Gene | Forward primer (5'→3') | Reverse primer (5'→3') | Annealing temperature (°C) | Product size (bp) |
|------|------------------------|------------------------|----------------------------|-------------------|
| GAPDH | GTAGTGAAGGCTGCTGCTGATG | CAAAGGTGGAGGAATGGCTGTC | 60°C | 131 |
| TPM2 | TGGAACTCCAGGAAATGCAACT | ATGAGGGATTTGAGGCTCTGGT | 60°C | 202 |
| CREB5 | CAAAGATGAACTTGGAGCAGGAG | AGAAAGCGTGTCGGTGTAGGAG | 60°C | 181 |
| GIPC2 | TTAGGCTGGAGTGTTTGTTTGGTAG | TGGATTGGTGGAGTCTTTAGTCATTTC | 60°C | 182 |
| DUSP1 | CAAGGGAGGTTACGAAGCGT | CCTGGTCGTACAGAGGGGTA | 60°C | 181 |

in the MH group were selected for metabolite extraction. The metabolite extraction mainly includes the following steps: following the addition of extraction solvent (methanol: acetonitrile: water, 2:2:1) and magnetic beads, samples underwent mechanical grinding, ultrasonic treatment, static centrifugation, and subsequent vacuum drying of the supernatant [26]. Subsequently, the reconstituted extracts were analyzed using an integrated UPLC-QTOF-MS system (Waters Acquity I-Class PLUS coupled with Waters Xevo G2-XS QTof). The column used is purchased from Waters Acquity UPLC HSS T3 column (1.8um 2.1*100 mm). The collision energy was set to 2V for low energy and ranged from 10 to 40V for high energy. The scanning frequency was 0.2 seconds per mass spectrum, with a full scan range of m/z 50–1200. The capillary voltage was maintained at 2500V in positive ion mode and -2000V in negative ion mode. The gradient elution procedure was detailed in Table 2. For both positive and negative ion modes, mobile phase A consisted of a 0.1% formic acid aqueous solution, and mobile phase B consisted of 0.1% formic acid acetonitrile. Injection volume 1 μL. Chromatographic separation and mass spectrometric detection were performed under optimized analytical conditions. Acquired data (MassLynx V4.2) were processed through Progenesis QI software for feature extraction, peak alignment, and data normalization. Metabolite identification was achieved by cross-referencing with the METLIN database, public repositories, and a custom-established Baimaike database, complemented by theoretical fragmentation pattern analysis. Mass accuracy thresholds were maintained at 100 ppm for precursor ions and 50 ppm for product ions. The PCA analysis and orthogonal partial least squares discriminant analysis (OPLS-DA) were conducted using the BMKCloud (https://international.biocloud.net/zh/dashboard). Differential metabolite screening was performed based on multivariate and univariate statistical analyses. The Variable Importance in Projection (VIP) values derived from the OPLS-DA model were initially employed to identify potential metabolic differences between cultivars or tissues. Subsequently, these findings were integrated with univariate analysis results to enhance the reliability of differential metabolite selection. The screening criteria were set at FC > 1, P < 0.05, and VIP > 1. The final results were visualized using a volcano plot. The cluster profile package was employed to perform an enrichment analysis of KEGG annotation results using the hypergeometric test, and an enrichment dot plot was created.

### Integrative investigation of transcriptional and metabolic networks

Comprehensive bioinformatics analysis of gene expression and metabolic profiles was carried out utilizing the BMK Cloud analytical suite (www.biocloud.net), including library quality assessment, principal component analysis, differential expression analysis, gene function and metabolic pathway annotation and functional enrichment, and combined analysis.

Weighted gene co-expression network analysis (WGCNA) was performed using the BMK Cloud analytical suite (www.biocloud.net) following the standard workflow. Based on transcriptome and metabolome data obtained from the platform's early-stage analysis, the differential expression analysis was performed using DESeq2_EBSeq with an FDR threshold of 0.05 and a fold change cutoff of 1.5. Analyze according to the previous groupings of MH and ML.

**Table 2. Gradient elution program used for chromatographic separation.**

| Time (min) | Flow rate (µL/min) | Mobile phase A (%) | Mobile phase B (%) |
|---|---|---|---|
| 0.0 | 400 | 98 | 2 |
| 0.25 | 400 | 98 | 2 |
| 10.0 | 400 | 2 | 98 |
| 13.0 | 400 | 2 | 98 |
| 13.1 | 400 | 98 | 2 |
| 15.0 | 400 | 98 | 2 |

(1) Mobile phase A: 0.1% formic acid aqueous solution.

(2) Mobile phase B: 0.1% formic acid acetonitrile.

## Results

### Comparison of melanin content in breast muscle

A comparison of high and low levels of melanogenesis and subsequent deposition in the breast muscle of Muchuan Black Bones Chickens was shown in Fig 1(A). As illustrated in Fig 1(B), black bones chickens were selected for the ML group (low melanin expression group) and the MH group (high melanin expression group) based on their melanin content; the expression levels of melanin in breast muscle were (103.193±3.598) mg/mL and (121.731±3.69) mg/mL, respectively. Statistical analysis revealed a significantly elevated melanin concentration in the pectoral muscle tissue of the high-level group compared with the low-level group *(P<0.01)*.

### Transcriptome analysis

**Raw sequencing data quality assessment summary.** Table 3 present a summary of the RNA-Seq results of Muchuan Black Bones chicken breast samples, comprising five ML and five MH samples, along with both raw and clean reads. The Q30 value exceeded 90.43%, and the Q20 value exceeded 96.11%, while the GC content ranged from 52.36% to 53.38%.

**Gene differential expression analysis.** Transcriptomic analysis revealed significant differential gene expression patterns between high and low melanin groups, identifying 73 DEGs (54 up-regulated and 19 down-regulated), as illustrated in Fig 1(C). The top ten DEGs based on $\log_2$(FC) were presented in Table 4.

**Data quality assessment of RT-qPCR.** The expression patterns of the two differentially expressed genes, validated by both transcriptome sequencing and RT-qPCR, were presented. The figure shown that the RNA-seq expression level of the *GIPC2* gene in the MH group was approximately 5-fold higher than that in the ML group, while the RT-qPCR expression level in the MH group was 4.5-fold higher in Fig 1(D). Similarly, for the *CREB5* gene (Fig 1(E)), the RNA-seq expression level in the MH group was 5-fold higher compared to the ML group, and the RT-qPCR expression level was 3.1-fold higher. These results were demonstrated a largely consistent expression pattern between the two methods. The transcriptional profiles of the *CREB* and the *GIPC* gene families obtained from RNA sequencing were presented in Fig 1(F). Transcriptome sequencing results revealed that, apart from the previously identified members, other *CREB* family members in the MH group did not exhibit consistent up-regulation. Within the *GIPC* family, *GIPC2* expression was elevated in the MH group compared to the ML group, whereas *GIPC1* expression levels remained comparable between the two groups.

**GO enrichment analysis of DEGs.** The classification of differentially expressed genes (DEGs) from the high and low melanin content groups into GO terms were revealed significant enrichment *(P<0.05)* across 6 functional categories. Biological processes (BP) were included auditory receptor cell stereocilium organization and axon development. Cellular

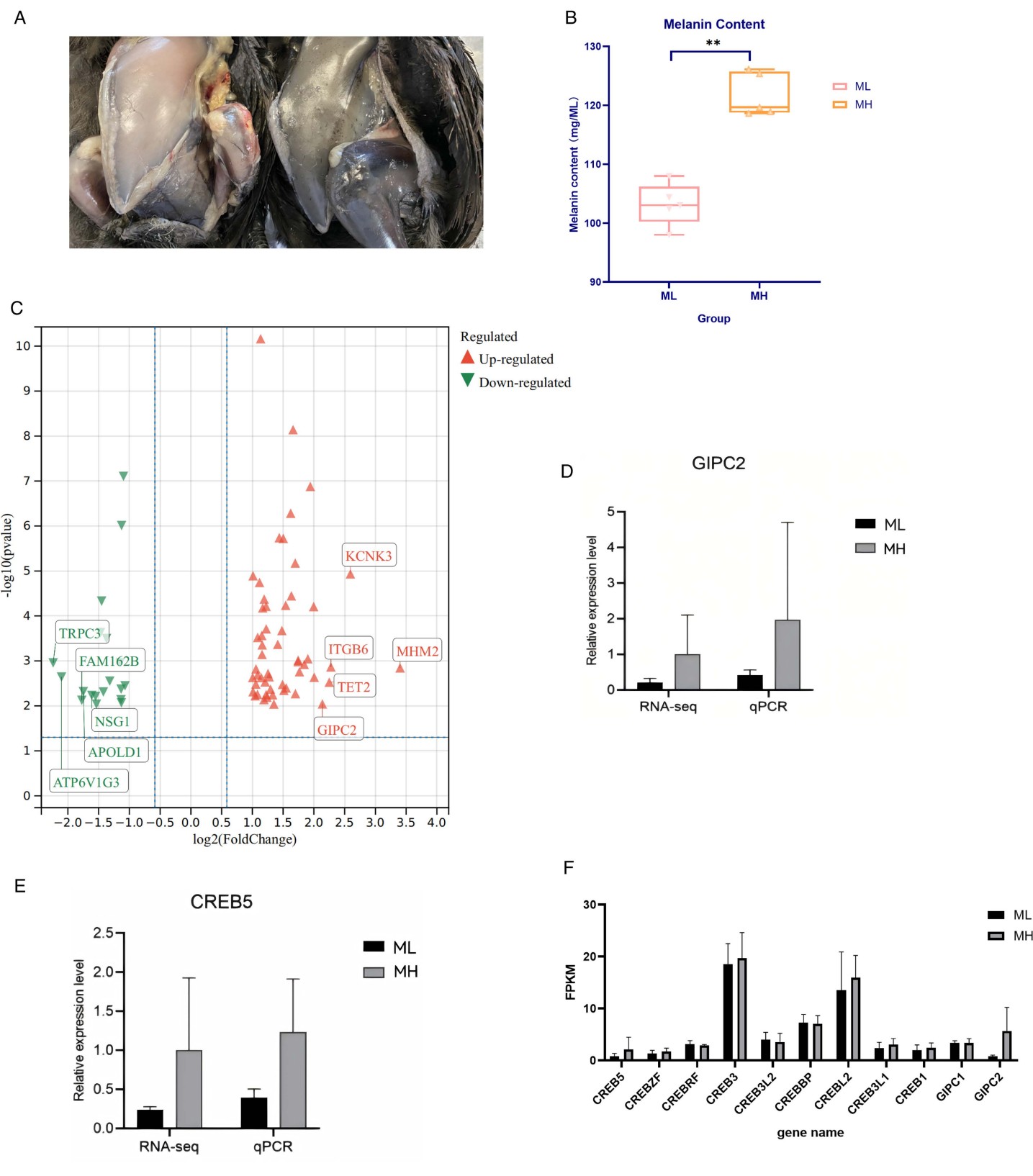

**Fig 1. The results of phenotype of Muchuan black bones chickens and the expression of genes. (A)** Comparison of blackness between MH and ML black bones chickens. **(B)** Statistical significance of melanin content. ** was meant the extremely significant. **(C)** DEGs volcano map. **(D)** Expression trend of differential gene transcriptome sequencing and real-time fluorescence quantitative PCR of *GIPC2*. **(E)** Expression trend of differential gene transcriptome sequencing and real-time fluorescence quantitative PCR of *CREB5*. **(F)** The transcriptome sequencing results of the *GIPC* gene family and the *CREB* gene family.

**Table 3. Summary statistics of sequence quality and comparison information.**

| Samples | Clean reads | Clean bases | GC Content | % ≥ Q30 | % ≥ Q20 |
|---|---|---|---|---|---|
| ML1 | 20,315,207 | 6,078,541,552 | 52.83% | 91.26% | 96.61% |
| ML2 | 21,641,148 | 6,474,947,590 | 53.05% | 91.99% | 96.89% |
| ML3 | 23,440,977 | 7,011,163,862 | 52.42% | 92.62% | 97.17% |
| ML4 | 19,689,063 | 5,890,581,038 | 52.93% | 91.99% | 96.89% |
| ML5 | 21,212,590 | 6,345,204,430 | 52.77% | 92.00% | 96.88% |
| MH1 | 22,064,781 | 6,598,044,174 | 53.38% | 94.93% | 98.01% |
| MH2 | 22,248,260 | 6,656,734,114 | 52.36% | 92.98% | 97.33% |
| MH3 | 22,280,888 | 6,665,761,316 | 52.64% | 92.48% | 97.15% |
| MH4 | 21,542,599 | 6,445,733,214 | 52.96% | 90.43% | 96.11% |
| MH5 | 20,685,551 | 6,187,410,710 | 52.37% | 91.56% | 96.70% |

(1) Samples: sample analysis number.

(2) Clean reads: Total paired-end reads post-quality filtration.

(3) Clean bases: Clean Data total bases.

(4) GC content: GC composition ratio in quality-filtered sequencing data.

(5) ≥Q30%: Percentage of high-quality bases (Q ≥ 30) in processed sequencing reads.

(6) ≥Q20%: Percentage of high-quality bases (Q ≥ 20) in processed sequencing reads.

**Table 4. The top ten genes that differential expressed.**

| Gene | Full name | $\log_2$(FC) | UP/DOWN |
|---|---|---|---|
| *MHM2* | male hypermethylated region 2 | 3.03 | UP |
| *KCNK3* | Potassium Two Pore Domain Channel Subfamily K Member 3 | 2.08 | UP |
| *ITGB6* | integrin subunit beta 6 | 2.15 | UP |
| *TET2* | Ten-Eleven Translocation 2 | 2.25 | UP |
| *GIPC2* | GIPC PDZ domain-containing family member 2 | 2.58 | UP |
| *MYOZ2* | Myozenin 2 | 2 | UP |
| *CREB5* | cAMP-responsive element-binding protein 5 | 1.37 | UP |
| *TMEM8C* | Transmembrane Protein 8C | 1.01 | UP |
| *TRPC3* | Transient Receptor Potential Canonical 3 | −2.24 | DOWN |
| *ATP6V1G3* | ATPase H+ transporting V1 subunit G3 | −2.1 | DOWN |

components (CC) were encompassed dendrites, integral plasma membrane components, and plasma membranes. Molecular functions (MF) were characterized by integrin binding, as illustrated in Fig 2(A).

**KEGG enrichment analysis of DEGs.** The KEGG pathway enrichment analysis was identified two metabolic pathways with significant enrichment *(P<0.05)*. The neuroactive ligand-receptor interaction pathway may regulate melanocyte activity via neuroendocrine signals, such as α-MSH/MC1R, thereby modulating melanin deposition in the pectoral muscles. The arginine-proline metabolism pathway may influence melanin production by supplying amino acid

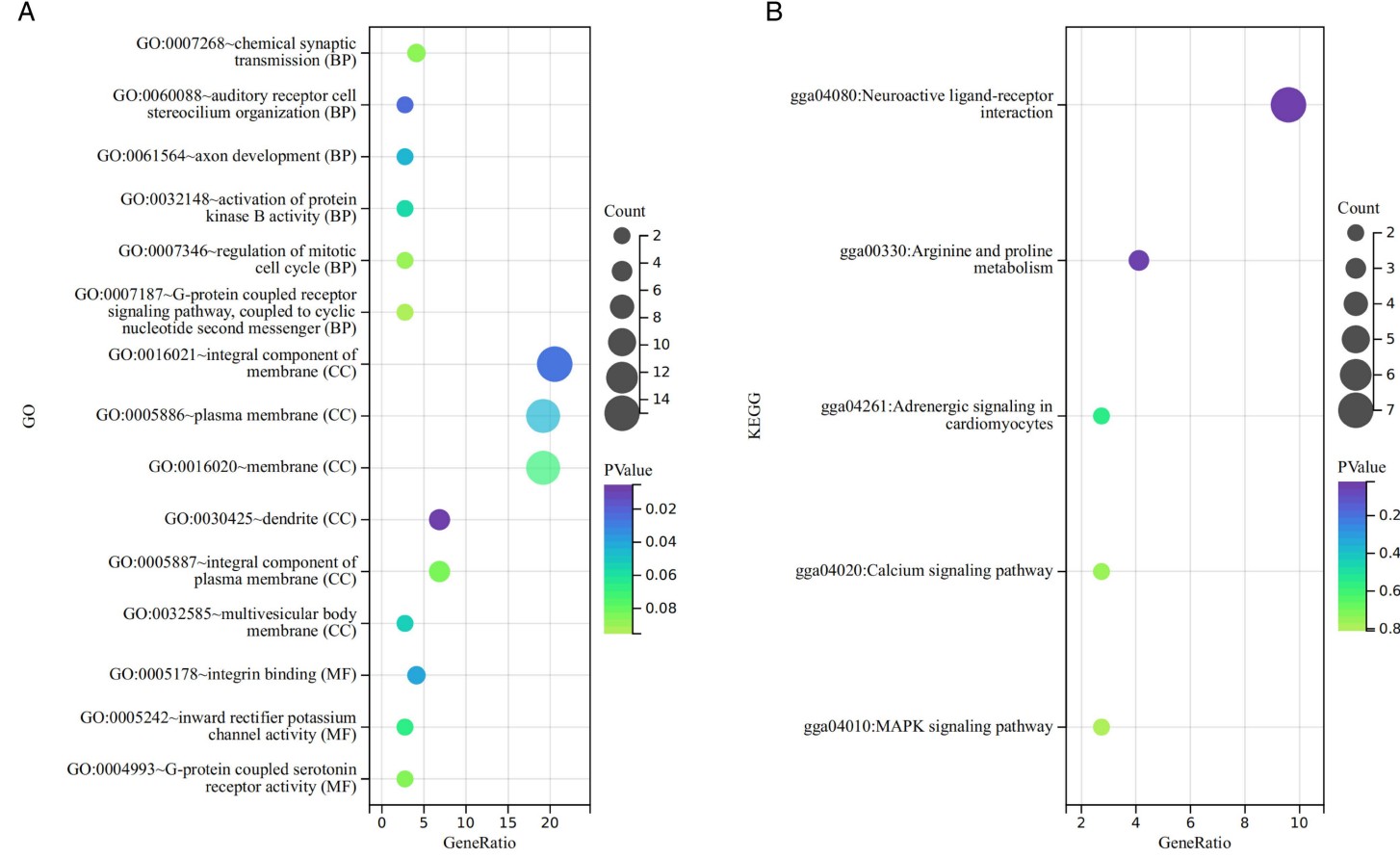

**Fig 2. Enrichment analysis of DEGs. (A)** Functional enrichment of DEGs GO. **(B)** Enrichment of DEGs KEGG pathway.

precursors and supporting the metabolic environment required for melanin synthesis, as demonstrated in Fig 2(B). Pathways associated with pigmentation were included adrenergic signaling in cardiomyocytes, the calcium signaling pathway, and the mitogen-activated protein kinase (MAPK) signaling pathway. Based on the enrichment results from both GO and KEGG, the adrenergic signaling pathway in cardiomyocytes included the *tropomyosin 2* (*TPM2*) gene (log$_2$(FC)=2.67) and *cAMP-responsive element binding protein 5* (*CREB5*) gene (log$_2$(FC)=1.37); the calcium signaling pathway included *histamine receptor H2* (*HRH2*) gene (log$_2$(FC)=1.09) and *fibroblast growth factor 1* (*FGF1*) gene (log$_2$(FC)=0.06); while the MAPK signaling pathway included *dual specificity phosphatase 1* (*DUSP1*) gene (log$_2$(FC)=0.34) and *fibroblast growth factor 1* (*FGF1*) gene (log$_2$(FC)=0.06). Differential gene expression ratios and regulation were shown in Table 5.

## Metabolome analysis

**PCA analysis.** The distribution of samples along PC1 and PC2 was illustrated in Fig 3, from which it was inferred that the MH and ML groups exhibited significant differences in metabolic composition.

**OPLS-DA analysis.** To enhance the identification of differential metabolites, this study also performed orthogonal partial least squares discriminant analysis (OPLS-DA). As illustrated in Fig 4(A), samples from the MH and ML groups were distinctly separable along the t1 axis. To mitigate overfitting, the model was verified in this experiment, as shown in

Table 5. The differentially expressed genes enriched by KEGG.

| Gene ID | Full name | log$_2$(FC) | UP/DOWN |
|---------|-----------|-------------|---------|
| TPM2 | tropomyosin 2 | 2.67 | Up |
| CREB5 | cAMP responsive element binding protein 5 | 1.37 | Up |
| HRH2 | histamine receptor H2 | 1.09 | Up |
| FGF1 | fibroblast growth factor 1 | 0.06 | Up |
| DUSP1 | dual specificity phosphatase 1 | 0.34 | Up |

Fig 4(B). The figure demonstrates that R2Y values predominantly cluster around 1, indicating that the model was robust and not subject to overfitting.

**Differential metabolite analysis.** The metabolomic screening identified 234 differentially abundant metabolites, among which 158 were significantly down-regulated and 76 were up-regulated, as shown in Fig 5(A). Among the down-regulated metabolites highlighted, several notable compounds include products associated with sugar metabolism, included uridine diphosphate glucuronic acid (UDP-glucuronic acid) and UDP-4-dehydro-6-deoxy-D-glucose. Compounds involved in the phospholipid synthesis pathway included CDP-ethanolamine. Metabolites related to redox and signal regulation included methyl gallate and glucose isoflavone.

**KEGG enrichment analysis of differential metabolites.** Significant enrichment of histidine metabolism, terpenoid backbone biosynthesis, and glycerophospholipid metabolism pathways was observed, as shown in Fig 5(B). The process of melanin synthesis was accompanied by the production of reactive oxygen species (ROS). The enhanced metabolism of histidine [27,28]. Terpenoid backbone biosynthesis provides the isoprenoid precursors required for the synthesis of ubiquinone (coenzyme Q), a key antioxidant molecule that participates in maintaining redox homeostasis during melanogenesis [29,30].

### Integrated transcriptome and metabolome analysis

**Weighted gene co-expression network analysis (WGCNA).** This study conducted WGCNA analysis on the transcriptome and metabolome data independently, revealing that certain modules, such as MEdarkmagenta and GEmagenta, exhibited strong positive correlations in the heatmap, as shown in Fig 6 Conversely, a strong negative correlation was observed between MElightyellow and GEdarkgreen.

**Bidirectional orthogonal partial least squares analysis.** Joint Score 1 accounts for the maximum shared variation between transcriptomic and metabolomic data, whereas Joint Score 2 captures the secondary shared variation. The distribution of W8 (red) was relatively close to that of B8 (blue), as illustrated in Fig 7(A).

Based on the results of element loading values, this study identified the top 15 differential genes and metabolites with the highest loading values in the first two principal components and constructed bar charts to illustrate those with the strongest associations, as shown in Fig 7(B), genes exhibiting significant upregulation, such as *TANGO2* (*transport and golgi organization 2 homolog*) and *NewGene_4870*, were listed, along with genes showing significant downregulation, included *CD63* (*CD63 molecule*) and *LARP6* (*La ribonucleoprotein 6, translational regulator*); genes such as *TPM2, CREB5, HRH5, FGF1*, and *DUSP1*, the expression levels of these genes were relatively low, as indicated by their small loading values ($|Loading_1| < 0.01$ and $|Loading_2| < 0.01$). Specific compounds, such as Semivioxanthin and Gossypol, demonstrate significant positive or negative loading values within the model, highlighting their crucial roles in metabolomic data.

### Discussion

The deposition of melanin in Muchuan Black Bones chickens has a pronounced impact on appearance quality, and higher levels of melanin are positively associated with improved nutritional value [31]. Although progress has been made

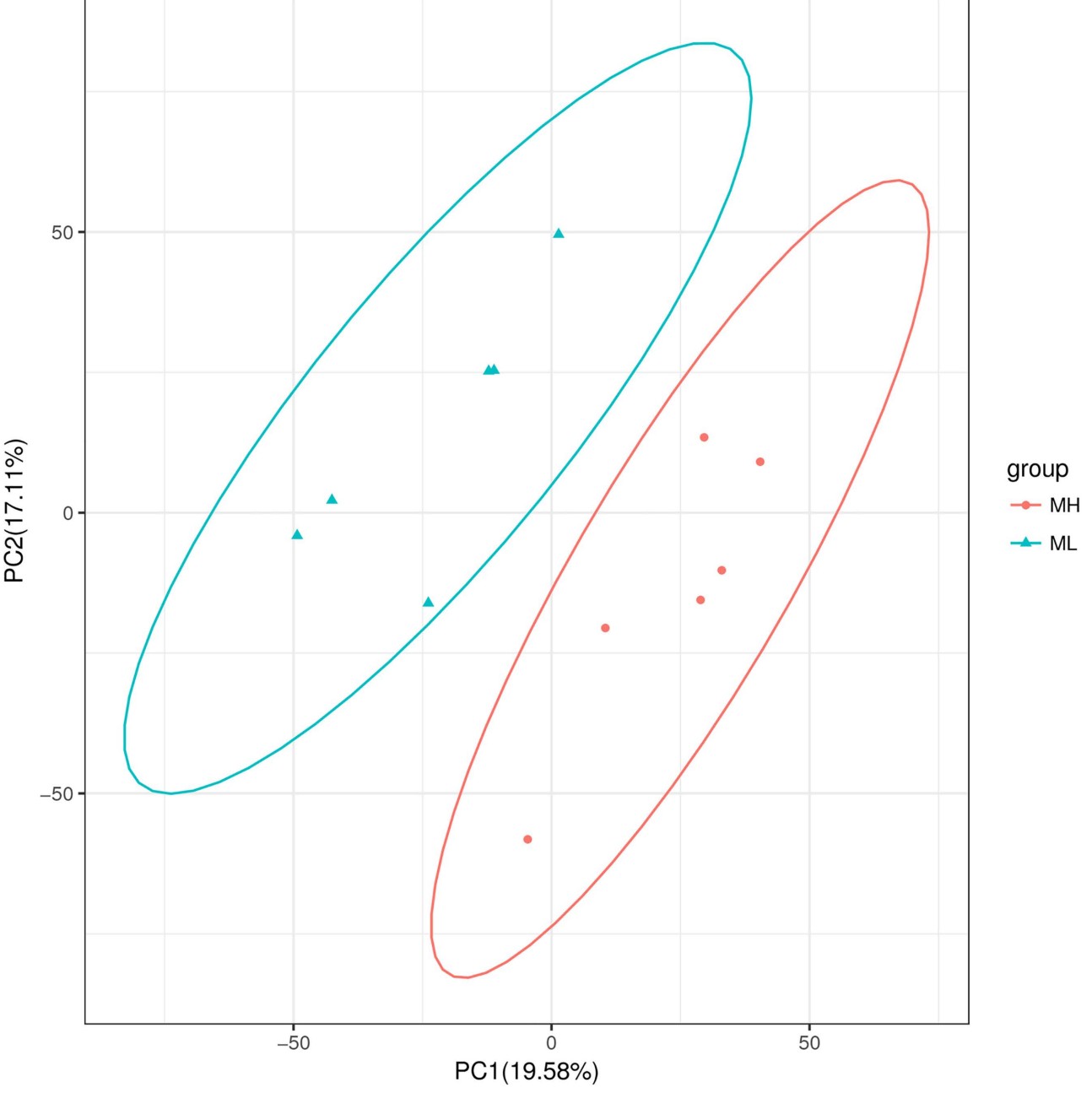

**Fig 3. PCA analysis of all samples in two dimensions.**

in breeding research on Muchuan Black Bones Chickens in recent years, most existing studies have primarily focused on selecting for black skin traits, emphasizing external appearance and market value. In contrast, research on melanin deposition in the breast muscle remains limited, and available data are largely derived from older birds [32]. Consequently, the specific influence of breast muscle melanin deposition on appearance-related traits—particularly its regulatory mechanisms across different growth stages—remains insufficiently explored. Through the previous observation, it was found that

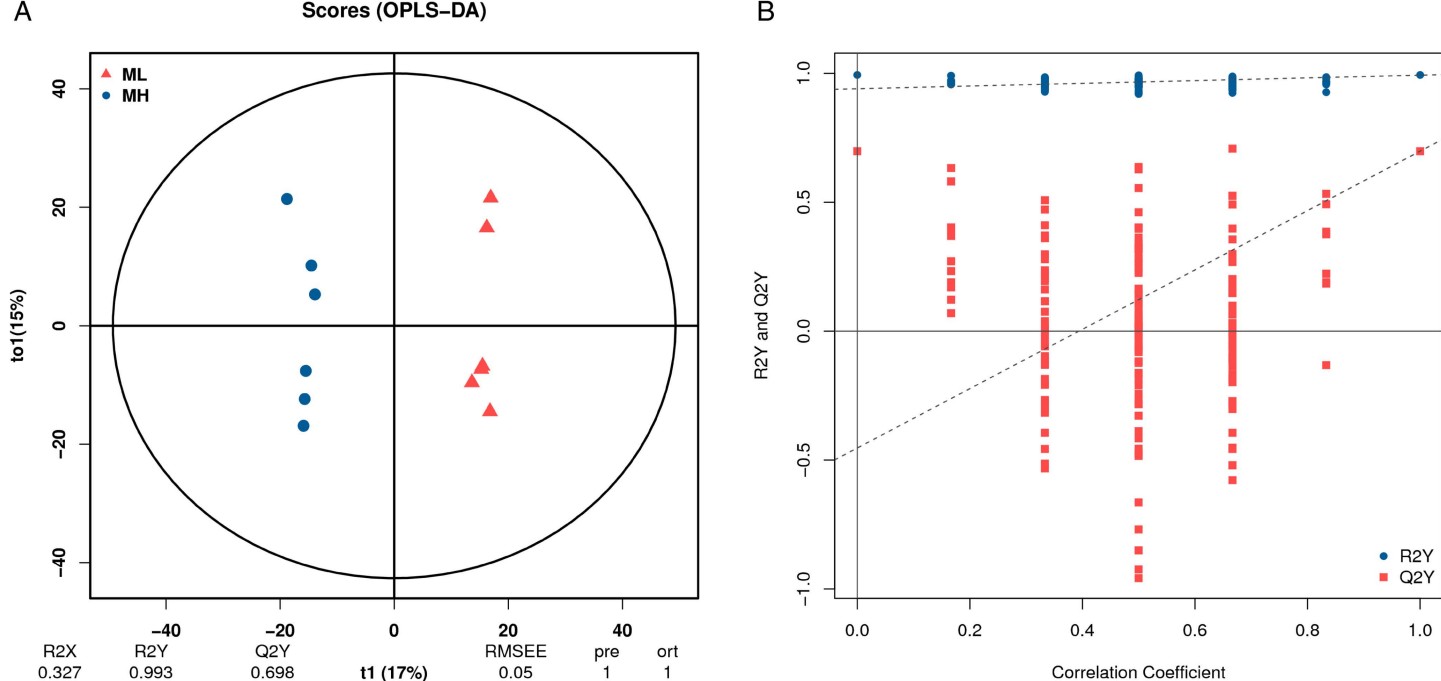

**Fig 4. OPLS-DA analysis of differential metabolites. (A)** Orthogonal Partial Least Squares Discriminant Analysis (OPLS-DA). **(B)** OPLS-DA model replacement test diagram.

the melanin deposition in the early growth stage (35 days of age) had shown a significant degree of separation, especially in the breast muscle and skin [33]. These findings highlight the necessity of further investigation into early-stage melanin deposition, with particular emphasis on developing breeding strategies that could optimize its contribution to appearance quality.

The transcriptomic analysis revealed that two genes associated with melanin synthesis, namely *GIPC2* and *CREB5*, were significantly up-regulated. Previous studies have demonstrated that the missense mutations D125N and E288K in *GIPC2* were implicated in malignant melanoma development [34]. As a transcription factor belonging to the *CREB* family, *CREB5* exerts its regulatory function by specifically interacting with cAMP response elements (CREs) in downstream genes [35], thereby playing a regulatory role in various cell-specific biological activities, including cell growth, survival, metabolism, and differentiation. It has been established that members of the *CREB* family can enhance melanogenesis through microphthalmia-associated transcription factor (*MITF*) activation; however, the specific role of *CREB5* in this context remains inadequately characterized. As an integral component of the *CREB* family, *CREB5* may indirectly influence *MITF* expression via analogous mechanisms [11]. A critical regulatory pathway for melanogenesis was the cAMP response elements (cAMP-PKA) signaling cascade, which can be further modulated by activating *CREB* family transcription factors. Consequently, *CREB5* may play a significant role in this pathway and impact the expression of downstream genes associated with melanogenesis. The RNA-seq results suggest that transcript-level compensation of *CREB5* by other *CREB* family members was minimal or absent. Although some family members were highly expressed, they did not display a uniform inverse pattern relative to *CREB5* that would support compensatory transcriptional responses. Although *GIPC2* exhibited an up-regulated trend in the MH group, no corresponding inverse expression pattern or negative correlation was observed with other family members. Therefore, current evidence supports only a potential transcriptional-level response for *GIPC2*, rather than confirming a definitive gene or messenger ribonucleic acid (mRNA) compensatory effect.

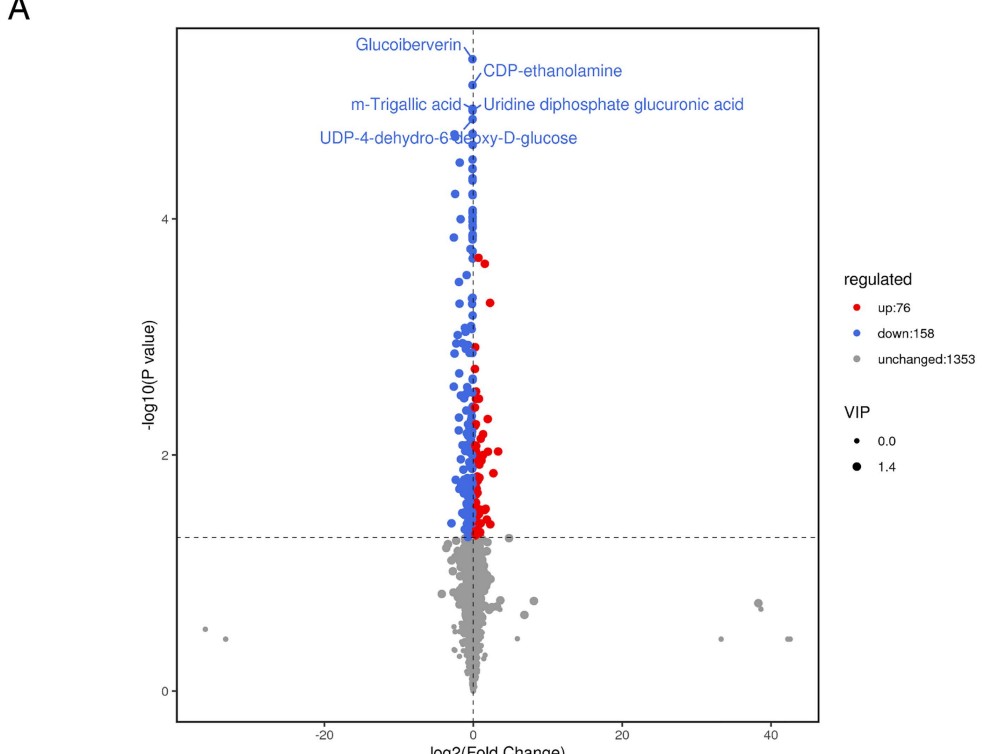

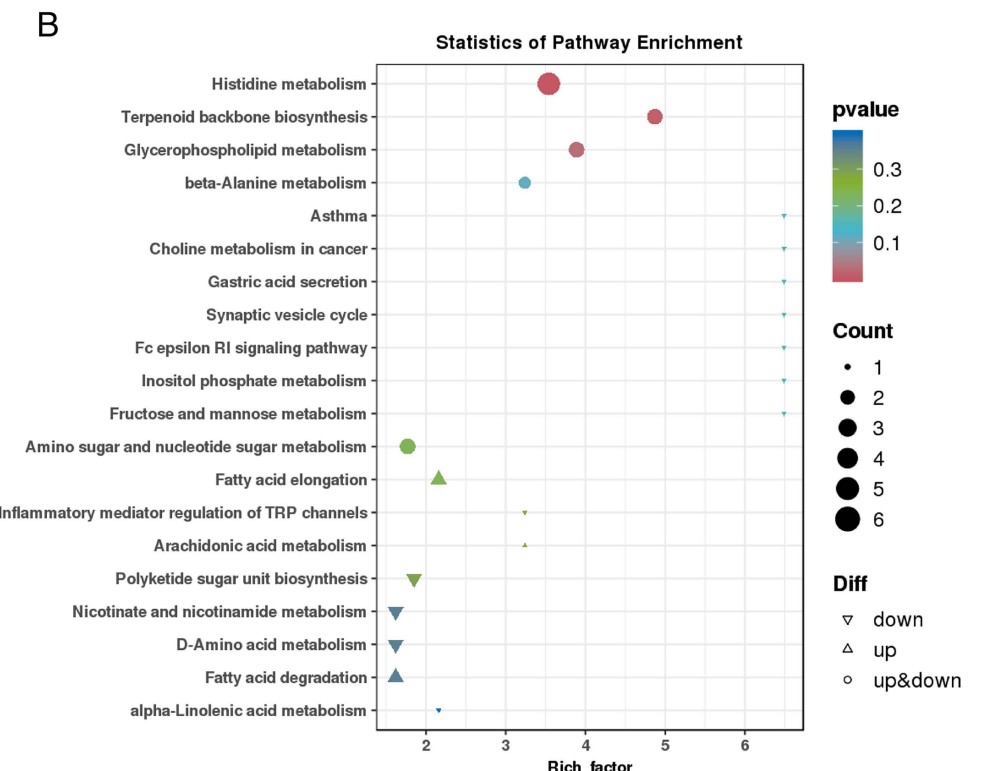

**Fig 5. The volcanic map of differential metabolites and KEGG enrichment map of differential metabolites. (A)** Volcanic map of differential metabolites. **(B)** KEGG enrichment map of differential metabolites.

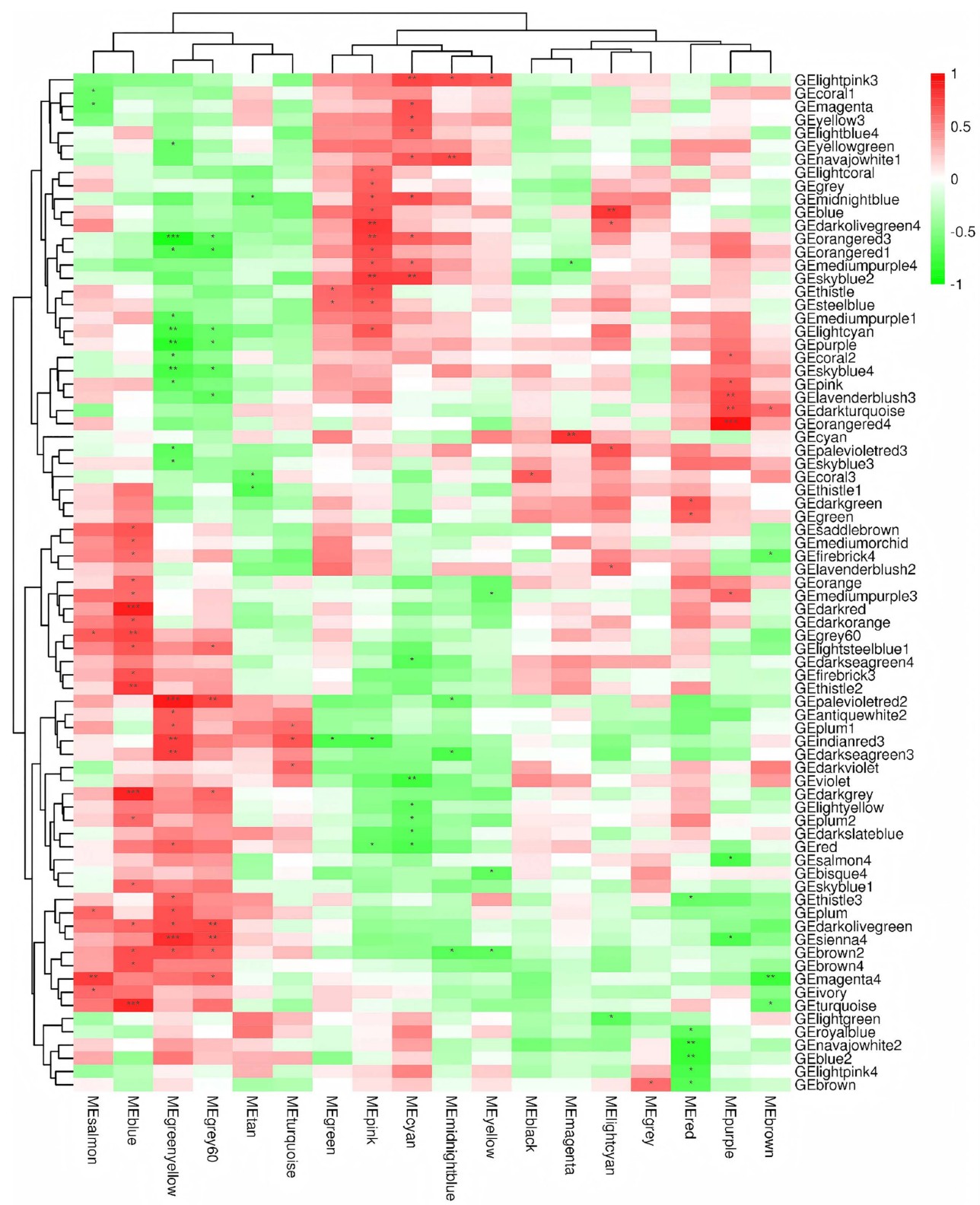

**Fig 6. Weighted gene Co-expression network analysis (WGCNA).**

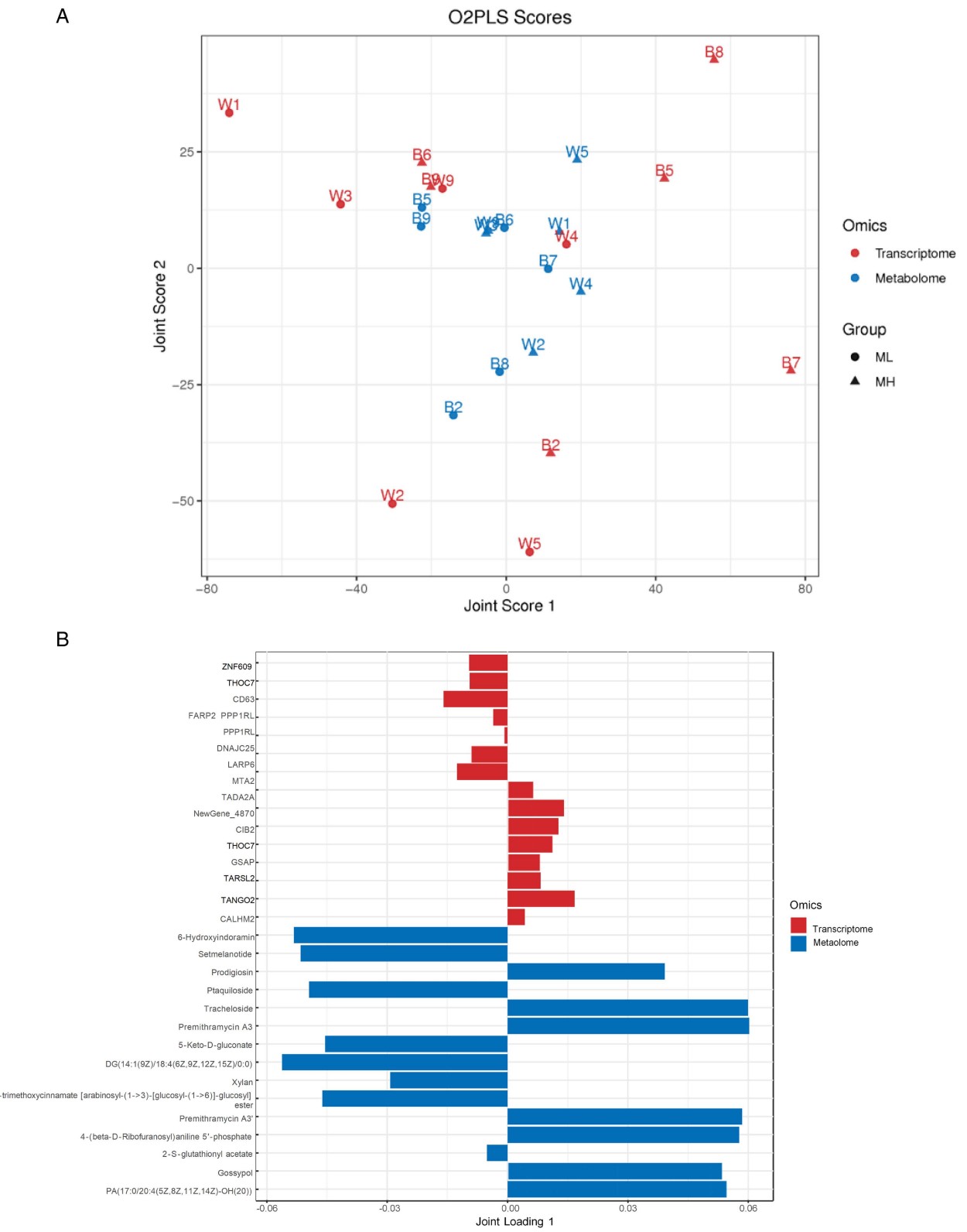

**Fig 7. Bidirectional orthogonal partial least squares analysis. (A)** ML vs MH O2PLS scores. **(B)** ML vs MH top15 O2PLS loadings bar.

In the GO enrichment analysis, integrins were critical adhesion receptors that regulate cell migration and attachment to the extracellular matrix. During development, melanocytes are migrate from the neural crest and successfully colonize target tissues such as the dermis or skeletal muscle [36]. The presence of this pathway suggest that integrin-mediated adhesion may facilitate melanocyte migration and anchoring within the pectoral muscle of Muchuan black-bone chickens, thereby promoting stable melanocyte settlement and subsequent melanin deposition.

In the KEGG enrichment pathways, the enrichment of the neuroactive ligand-receptor interaction pathway was highly consistent with the regulatory mechanisms of melanogenesis. Melanin production in melanocytes was under neuroendocrine control, and a well-known example was the melanocortin 1 receptor (MC1R), a neuroendocrine receptor activated by ligands such as α-MSH. MC1R signaling initiates the cAMP pathway, thereby promoting eumelanin synthesis [37]. The enrichment of this pathway in Muchuan black-bone chickens, therefore, highlights the plausible involvement of neuroendocrine factors in the regulation of melanin deposition in skeletal muscle. The arginine–glutamate metabolism pathway provides metabolic energy and biosynthetic precursors to support melanocyte activity [38]. Moreover, this pathway cooperates with oncogenic signals such as BRAF to drive metabolic reprogramming in melanocytes [39]. Importantly, targeting key enzymes within this pathway, for example, glutaminase inhibition, has been shown to reduce melanin production and suppress melanoma progression, underscoring its dual role in pigment regulation and tumor biology [40]. In the differential metabolite analysis, this study identified Glucoiberverin and Uridine diphosphate glucuronic acid (UDP glucuronic acid) as key metabolites that were significantly associated with melanin deposition. These metabolites were primarily linked to cellular metabolism, which played a crucial role in regulating various biochemical processes that could influence melanin accumulation. Glucoiberin was a natural sulfur-containing glucosinolate that could be enzymatically hydrolyzed into sulforaphane, a compound known to reduce cell viability and modulate the expression levels of various histone deacetylases. Through this mechanism, sulforaphane regulates the acetylation and methylation of specific lysine residues on histones H3 and H4 in malignant melanoma cells [41]. While neither Glucoiberverin nor UDP-glucuronic acid played a crucial role in key melanogenic processes, particularly in facilitating the bioconversion of tyrosine to melanin pigments, their contributions to cellular energy metabolism and metabolic regulation were vital for the overall process of melanin deposition. UDP-glucuronic acid was a key player in glycosylation reactions, particularly in the biosynthesis of glycosaminoglycans and the detoxification processes within the cell [42]. This metabolite facilitated the conjugation of various compounds, enhancing their solubility and excretion, and thereby indirectly influencing cellular function. Its role in glycosylation also extended to the modification of proteins and lipids that may be involved in the regulation of melanin-producing enzymes or the structural integrity of melanocytes. UDP-glucuronic acid may influence the glycosylation of melanogenic enzymes or cellular receptors, thereby affecting the efficiency and regulation of melanin synthesis and ultimately modulating melanin deposition in tissues [43]. Although these metabolites do not directly participate in the melanogenic biochemical pathway, they play pivotal roles in modulating cellular homeostasis, maintaining energy metabolism, and regulating protein post-translational modifications, thereby exerting a substantial influence on melanin deposition processes. This finding highlight the important role of metabolic intermediates in regulating pigment biosynthesis and offered new avenues for exploring the complex interactions between metabolic networks and phenotypic pigmentation, particularly in economically important poultry such as Muchuan Black Bones chickens. These indirect roles may also offer insights into improving the nutritional value and quality of appearance based on melanin content. In addition, differential metabolomic analysis revealed significant enrichment of metabolites such as CDP-ethanolamine, m-trigallic acid, and UDP-4-dehydro-6-deoxy-D-glucose. CDP-ethanolamine (cytidine diphosphate ethanolamine) was a key intermediate in the Kennedy pathway, which was primarily involved in the biosynthesis of phosphatidylethanolamine (PE) [44]. Membrane fluidity was crucial in the process of melanosome formation: it promoted membrane bending and vesicle formation, influenced the localization and function of membrane proteins, and maintained material transport between organelles, thereby ensuring the normal maturation of melanosomes and melanin synthesis. CDP-ethanolamine, as a precursor for PE synthesis, indirectly affected membrane fluidity by regulating PE levels, and its abnormality might have been related to pigmentation disorders.

Information on m-trigallic acid was currently limited, and its biological role remains unclear, necessitating further investigation to determine its potential physiological significance in animal systems. UDP-4-dehydro-6-deoxy-D-glucose was a sugar-nucleotide intermediate involved in the biosynthesis of deoxy sugars in microorganisms and plants. Although its function in animals was not well characterized, its enrichment in this study suggests potential metabolic relevance. However, their significant differential abundance in this study suggest they may participate in melanin-related metabolic regulation, warranting further mechanistic investigation.

According to the metabolomics, the KEGG was analyzed for two pathways: phenylalanine, tyrosine, and tryptophan biosynthesis. Phenylalanine was enzymatically converted to tyrosine by phenylalanine hydroxylase (PAH), after which tyrosine undergoes further oxidation to form dihydroxyphenylalanine (DOPA) through the action of tyrosinase [45]. Subsequently, DOPA was transformed into dopachrome via a series of intricate chemical reactions, ultimately yielding eumelanin or pheomelanin, which represent the primary types of melanin [46]. A substantial number of studies have identified key genes involved in melanin biosynthesis and pigmentation regulation, often referred to as "marker genes" in melanogenesis. These included *EDNRB2, EDN3, TYR, TYRP1, MITF,* and *PMEL17*, which played crucial roles in different stages of melanogenesis, ranging from melanoblast migration to enzymatic catalysis and melanosome formation. However, these genes were not detected in this study due to sequencing limitations; thus, low-expressed genes might not have been effectively captured. Additionally, external factors such as diet, light exposure, and temperature could have influenced melanin production, leading to changes in gene expression patterns. Although no typical genes regulating melanin were detected, this method identified new regulatory factors beyond the core pathway, such as *CERB5, GIPC2*, etc.

## Conclusion

This study identified that the *GIPC2* and *CREB5* genes were associated with melanin deposition. In parallel, several metabolites, including CDP-ethanolamine, m-trigallic acid, UDP-4-dehydro-6-deoxy-D-glucose, and UDP-glucuronic acid, were found to be significantly associated with pigmentation. These findings not only provided candidate targets for future multi-omics studies (e.g., epigenomics, proteomics) to unravel the regulatory mechanisms of pigmentation but also offered potential biomarkers for selective breeding in black-bone chickens.

## Author contributions

**Conceptualization:** Ma Zheng.

**Data curation:** Jie Hong-Wei, Yao Xiu-Mei.

**Funding acquisition:** Li Liang, Ye Fei.

**Investigation:** Yin Qiong.

**Project administration:** Gao Ming-Chao.

**Resources:** Gao Ming-chao.

**Supervision:** Xiang Hai, Zhang Zheng-Fen.

**Writing – original draft:** Liu Wei.

**Writing – review & editing:** Li Hua, Qi Hao, Ye Fei.

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
