## [Decision Letter · Decision Letter 0]

3 Sep 2025

Dear Dr. Fei,

Thank you for submitting your manuscript to PLOS ONE. After careful consideration, we feel that it has merit but does not fully meet PLOS ONE’s publication criteria as it currently stands. Therefore, we invite you to submit a revised version of the manuscript that addresses the points raised during the review process.

**ACADEMIC EDITOR:** Thank you for submitting your manuscript to PLOS ONE. We were able to secure only one reviewer for your paper, even though we invited many experts on behalf of your paper. After careful editorial consideration, and in view of the comments of the reviewer, we feel that your paper has merit but requires revision. Therefore, we invite you to submit a revised version of the manuscript that addresses comprehensively addresses the points of both the reviewer and the editor.

We look forward to receiving your revised manuscript.

Kind regards,

Andre van Wijnen

Academic Editor

PLOS ONE

Journal Requirements:

1. Please ensure that your manuscript meets PLOS ONE's style requirements, including those for file naming. The PLOS ONE style templates can be found at https://journals.plos.org/plosone/s/file?id=wjVg/PLOSOne_formatting_sample_main_body.pdf and https://journals.plos.org/plosone/s/file?id=ba62/PLOSOne_formatting_sample_title_authors_affiliations.pdf.

 [This research was funded by the Guangdong Basic and Applied Basic Research Foundation (2022A1515111078) and Guizhou Science and Technology Key Project ([2022] No. 034).].

6.  Thank you for stating the following in the Competing Interests section:

[Guangdong Tinoo’s Food Co., Ltd].  

We note that one or more of the authors are employed by a commercial company: name of commercial company.

Within your Competing Interests Statement, please confirm that this commercial affiliation does not alter your adherence to all PLOS ONE policies on sharing data and materials by including the following statement: ""This does not alter our adherence to  PLOS ONE policies on sharing data and materials.” (as detailed online in our guide for authors http://journals.plos.org/plosone/s/competing-interests) . If this adherence statement is not accurate and  there are restrictions on sharing of data and/or materials, please state these. Please note that we cannot proceed with consideration of your article until this information has been declared.

7. Please include a caption for figure 1 to 6 and table 1 and 2.

Additional Editor Comments:

Editorial Comments:

Please present Figure 1D with the same level of data transparency as Figure 1B. In addition, expand figure 1D by presentation of RNA-seq data for other CREB members (e.g., CREB1/2/3/4) and GIPC members (e.g., GIPC1, GIPC3) to assess whether there is the possibility for gene/mRNA compensation.

Reviewer #1:

Reviewers' comments:

Reviewer's Responses to Questions

**Comments to the Author**

1. Is the manuscript technically sound, and do the data support the conclusions?

Reviewer #1: Yes

2. Has the statistical analysis been performed appropriately and rigorously?

Reviewer #1: Yes

3. Have the authors made all data underlying the findings in their manuscript fully available?

Reviewer #1: Yes

4. Is the manuscript presented in an intelligible fashion and written in standard English?

Reviewer #1: Yes

Reviewer #1: This manuscript employs a multi-omics approach (transcriptomics and metabolomics) to investigate the molecular basis of variable melanin deposition in the breast muscle of 35-day-old Muchuan Black-Bone chickens. The study identifies novel candidate genes (CREB5, GIPC2) and metabolites beyond the well-characterized melanogenic pathway, providing new insights into a commercially important trait. However, the manuscript requires significant revisions to:

Abstract Discrepancy: A critical inconsistency exists where the abstract highlights "Glucose-6-phosphate" as a key metabolite, but this compound is absent from the main results and discussion, which focus on entirely different metabolites (e.g., CDP-ethanolamine, UDP-glucuronic acid). This must be corrected for accuracy.

Superficial Interpretation of Pathways: The authors list enriched pathways (e.g., Arginine-Proline metabolism, Histidine metabolism) but do not adequately discuss their plausible biological connection to melanogenesis. The manuscript would be significantly strengthened by deeper mechanistic hypothesizing.

Insufficient Methodological Detail: Key details are missing from the methods section, preventing full reproducibility. Most critically, the qPCR primer sequences and reference genes are omitted. The metabolomics extraction solvent and chromatography conditions are also not specified.

Underdeveloped Integration: While both omics datasets are presented, their true integration is limited to showing correlations between modules (WGCNA) and joint variation (O2PLS). The most valuable step—directly correlating key DEGs with key metabolites to generate testable hypotheses—is missing.

**Do you want your identity to be public for this peer review?** For information about this choice, including consent withdrawal, please see our Privacy Policy

Reviewer #1: No

---

## [Author Response · Author response to Decision Letter 1]

26 Oct 2025

Additional Editor Comments:

Editorial Comments:

Please present Figure 1D with the same level of data transparency as Figure 1B. In addition, expand figure 1D by presentation of RNA-seq data for other CREB members (e.g., CREB1/2/3/4) and GIPC members (e.g., GIPC1, GIPC3) to assess whether there is the possibility for gene/mRNA compensation.

Reviewer #1:Regarding the issues raised by the journal editors, corresponding changes have been made in the revised version of the article (the changed parts are highlighted in yellow) and in Fig1(D).

---

## [Decision Letter · Decision Letter 1]

9 Nov 2025

Dear Dr. Fei,

Thank you for submitting your manuscript to PLOS ONE. After careful consideration, we feel that it has merit but does not fully meet PLOS ONE’s publication criteria as it currently stands. Therefore, we invite you to submit a revised version of the manuscript that addresses the points raised during the review process.

We look forward to receiving your revised manuscript.

Kind regards,

Andre van Wijnen

Academic Editor

PLOS ONE

Journal Requirements:

Additional Editor Comments:

Editorial Comments: This paper is acceptable for publication pending final revisions that address minor comments.

Reviewers' comments:

Reviewer's Responses to Questions

**Comments to the Author**

Reviewer #1: All comments have been addressed

2. Is the manuscript technically sound, and do the data support the conclusions?

Reviewer #1: Yes

3. Has the statistical analysis been performed appropriately and rigorously?

Reviewer #1: Yes

4. Have the authors made all data underlying the findings in their manuscript fully available?

Reviewer #1: Yes

5. Is the manuscript presented in an intelligible fashion and written in standard English?

Reviewer #1: Yes

Reviewer #1: The manuscript presents a well-designed multi-omics study that "Integrated Transcriptome and Metabolome Analysis of Melanin-related genes and Metabolites in Muchuan Black Bones Chickens". The identification of novel candidate genes, and metabolites is a valuable contribution to the field. The revisions made in response to the previous editorial and reviewer comments have significantly improved the manuscript's clarity and depth.

However, several issues, primarily concerning language, clarity, and data interpretation, still need to be addressed before the manuscript is suitable for publication.

**Do you want your identity to be public for this peer review?** For information about this choice, including consent withdrawal, please see our Privacy Policy

Reviewer #1: **Yes: ** Owais Iqbal

---

## [Author Response · Author response to Decision Letter 2]

20 Nov 2025

Dear Editor,

We would like to sincerely thank you and the reviewers for the time and effort you have devoted to evaluating our manuscript entitled "Transcriptome and Metabolome Combined to Analyze the Key Genes and Metabolites Related to the Melanin Content of Muchuan Black Bones Chickens" (Manuscript ID:PONE-D-25-36595). We greatly appreciate the constructive comments and valuable suggestions, which have helped us to improve the quality of our work. We have carefully revised the manuscript according to the reviewers’ comments. Below, we provide a detailed point-by-point response to each comment.

Journal Requirements:

Response:We reviewed all cited references using the Retraction Watch Database, and no retracted publications were identified.

The manuscript presents a well-designed multi-omics study that "Integrated Transcriptome and Metabolome Analysis of Melanin-related genes and Metabolites in Muchuan Black Bones Chickens". The identification of novel candidate genes, and metabolites is a valuable contribution to the field. The revisions made in response to the previous editorial and reviewer comments have significantly improved the manuscript's clarity and depth.

However, several issues, primarily concerning language, clarity, and data interpretation, still need to be addressed before the manuscript is suitable for publication.

Response:We made revisions to the manuscript, improving the language expression, enhancing the clarity of the charts, and modifying the image sizes, in an effort to meet the requirements of the journal. However, changing the size of the image may affect some clarity.

Sincerely,

YE Fei

Foshan University

2025/11/19

---

## [Decision Letter · Decision Letter 2]

28 Nov 2025

Integrated Transcriptome and Metabolome Analysis of Melanin-related genes and Metabolites in Muchuan Black Bones Chickens

PONE-D-25-36595R2

Dear Dr. Fei,

We’re pleased to inform you that your manuscript has been judged scientifically suitable for publication and will be formally accepted for publication once it meets all outstanding technical requirements.

Kind regards,

Andre van Wijnen

Academic Editor

PLOS ONE

Additional Editor Comments (optional):

Reviewers' comments:

Reviewer's Responses to Questions

**Comments to the Author**

Reviewer #1: All comments have been addressed

2. Is the manuscript technically sound, and do the data support the conclusions?

Reviewer #1: Yes

3. Has the statistical analysis been performed appropriately and rigorously?

Reviewer #1: Yes

4. Have the authors made all data underlying the findings in their manuscript fully available?

Reviewer #1: Yes

5. Is the manuscript presented in an intelligible fashion and written in standard English?

Reviewer #1: Yes

Reviewer #1: There are some spelling, space and formatting issues in the manuscript. Please check and revised carefully

**Do you want your identity to be public for this peer review?** For information about this choice, including consent withdrawal, please see our Privacy Policy

Reviewer #1: **Yes: ** Owais Iqbal

---

## [Editor Report · Acceptance letter]

PONE-D-25-36595R2

PLOS One

Dear Dr. Fei,

I'm pleased to inform you that your manuscript has been deemed suitable for publication in PLOS One. Congratulations! Your manuscript is now being handed over to our production team.

Kind regards,

on behalf of

Dr. Andre van Wijnen

Academic Editor

PLOS One